# Advancing training effectiveness prediction in mass sport through longitudinal data: A mathematical model approach based on the Fitness-Fatigue Model

**Wenxing Wang**[ID][☉], **Yuanhui Zhao**[☉], **Xiao Hou, Wenlang Yu, Hong Ren**[ID]*

School of Sport Science, Beijing Sport University, Beijing, China

☉ These authors contributed equally to this work.
* renhong@bsu.edu.cn

## Abstract

Despite the critical need for scientific training load assessment in mass sports, the Fitness-Fatigue Model (FFM) requires further mathematical optimization and practical output indicators. The aim of this study was to optimize the mathematical relationship between "adaptation" and "fatigue" in the FFM, identify generalizable model output indicators, and evaluate its performance in predicting training effectiveness in mass sport. To account for the nonlinear and time-varying characteristics of training effectiveness, this study proposed new mathematical assumptions and optimized parameters against individual longitudinal data. The external load (speed and wattage) and internal load (wearable-compatible heart rate variability [HRV] and heart rate recovery [HRR] related indicators) of each training day were collected for 28–42 days per person (420 paired data from 13 subjects during 12 weeks of medium-intensity continuous cycling). The longitudinal data were used to perform parameter estimation and model evaluation for each individual separately. When the optimal model output indicator was selected, the $R^2$ values of the optimized model ranged from 0.61–0.95, with fitting root mean square error (RMSE) at 0.07–0.37, and mean absolute percentage error (MAPE) in predictive ability assessment at 3.99%−31.99%. However, a few individuals had larger fitting errors (minimum $R^2$ of 0.32, maximum RMSE of 0.90) and predictive errors (maximum MAPE of 86.57%) when the output indicator was inappropriate. The original model generally has lower $R^2$ and higher RMSE and MAPE. This shows the optimization of functional relationships and the application of individual longitudinal data have resulted in better performance of the model, but optimal indicator selection varies by individual. Furthermore, HRV and HRR related indicators are generalizable model output indicators that can be used to predict training effectiveness in mass sports through wearable devices and machine learning technology. However, the study has limitations including the homogeneous sample

**Data availability statement:** All relevant data are within the manuscript and its Supporting Information files.

**Funding:** This work was supported by the National Key R&D Program of China Foundation (2018YFC2000600); and the Fundamental Research Funds for Central Universities (2024016).

**Competing interests:** The authors have declared that no competing interests exist.

and single training type. Future research should validate the model across different sports and populations, investigating the factors affecting model fitting and prediction.

---

## 1 Introduction

Training load and its effectiveness evaluation play a crucial role in competitive and recreational sports. A reasonable training load is essential in athletic training [1–5], but studies have shown that coaches often struggle to develop optimal training plans without precise data feedback, leading to inconsistencies between planned and actual training loads [6]. In mass training, a well-designed training plan can enhance physical fitness and prevent chronic diseases [7–9] while improper training load arrangements can result in injuries [10]. Therefore, establishing a scientific and predictive method for assessing training effectiveness has high practical significance for the improvement of sports performance and health benefits.

The Fitness-Fatigue Model (FFM), proposed by Banister and his team, simplifies the training process and quantifies the training load and its effectiveness [11]. However, the mathematical relationship in the model needs optimization. Physiologically, "adaptation" represents the positive functional and structural changes induced by training stimuli, such as an increase in muscle protein synthesis [12], mitochondrial-associated indices [13], and neuromuscular transmission efficiency [14]. Conversely, "fatigue" represents a temporary decline in bodily function that occurs after exercise, which may be related to factors such as muscle glycogen depletion [15], accumulation of metabolites [16,17], and nervous system fatigue [18]. FFM posits that external loads cause adaptive and fatigue responses, changing over time [19,20]. These factors jointly determine the overall internal response. By mathematically relating "adaptation" and "fatigue", researchers develop training load prediction models [21]. These models have proven effective in predicting training effects and influenced competitive sports [22–28], such as their application in pre-competition tapering strategy. Researchers can compare the potential effects of different tapering plans based on FFM, reducing reliance on coach experience [20,26,29,30]. However, traditional FFM uses first-order linear equations to describe the relationship between training response and external loads [11,21]. This doesn't reflect the nonlinear exercise training response and fading of adaptation and fatigue over time, inconsistent with reality. Due to the decisive role of the mathematical relationship between "adaptation" and "fatigue" in the FFM fitting effect [25], it is necessary to optimize the mathematical relationship between models to increase its interpretability and model fitting effectiveness.

Furthermore, while FFM is predominantly used in competitive sports, some studies have shown good model fitting in the general population [24]. This suggests that FFM has potential in recreational sports, but there is insufficient research evidence to support its application in the general population. With the rapid development of wearable devices in recent years, ordinary individuals can now access a wealth of training data (such as exercise heart rate, average running speed, etc.) through sports bands, sports watches, and other devices. However, there is a lack of means to process and



utilize this data, making it difficult for ordinary individuals to receive suitable training plan adjustments. As people increasingly focus on daily fitness activities, those without professional coaching guidance may be at risk of sports injuries and a decline in their quality of life. Therefore, the use of data-based models to optimize the effects of exercise interventions has gained attention [31–33]. The mathematical structure of FFM allows its parameters to be estimated by machine learning algorithms [21]. With sufficient data accumulation, the model can provide personalized guidance based on individual characteristics, thereby enhancing sports performance and preventing sports injuries. However, research on the application of FFM in the general population is limited, and the model output indicators selected in existing studies are also not easily applicable to mass sports (using invasive indicators to evaluate training effectiveness in the daily exercise of ordinary individuals is unrealistic). Additionally, considering the varying physiological responses of individuals to the same external load level, the focus of training load assessment should be on the "personalized" evaluation of training effects and the subsequent adjustment of external loads [34,35]. FFM can obtain personalized parameters based on individual longitudinal data and machine learning techniques for personalized evaluation. But most studies have only established universal model parameters and have not explored the "personalized" advantages of this model.

Therefore, the objective of this study is to optimize the mathematical relationship between "adaptation" and "fatigue" of FFM, identify generalizable model output indicators, and evaluate the model's performance in predicting training effectiveness in mass sport. To validate the model, we chose moderate-intensity continuous training, a common training mode in recreational sports. Moreover, training load comprises both external and internal components [36]. External load refers to the predetermined physical stress imposed on an individual during training, such as exercise intensity and volume. Internal load, on the other hand, reflects the physiological responses of the individual to this external stress [37]. Accordingly, for model input and output indicators, we utilized the power bicycle to collect external load and selected heart rate variability (HRV) and heart rate recovery (HRR) related indicators that reflect autonomic nervous activity as internal load indicators. Paired data collected on each training day were used for parameter estimation and mathematical modeling for each individual.

We hypothesize that (1) the mathematical modeling in this study will optimize the relationship between adaptation and fatigue, improving model interpretability through physiologically meaningful parameters; (2) HRV and HRR will serve as applicable and generalizable model output indicators for mass sports; (3) The model will demonstrate superior performance to the original model fitting with the higher fitting effect and lower prediction error through longitudinal data. It will also be capable of offering personalized evaluation and prediction of training outcomes, thereby offering effective guidance for improving sports performance and physical fitness in the general population.

## 2 Methods

### 2.1 Optimization of mathematical models

Based on the nonlinear and time-varying characteristics of training effectiveness, this study proposed new mathematical assumptions and formulated them into an optimized mathematical model. The specific modeling assumption are as follows:

(1) During training, fatigue ($F$) and adaptation ($A$) changes are primarily driven by external loads, and there is no occurrence of overtraining. This study retains the core concept of the FFM originally proposed by Banister et al., which describes performance as a function of adaptation and fatigue resulting from external loads [11,20,19]. However, to avoid the complexity associated with overtraining responses, this assumption is added to ensure the model focuses on the typical training response.

(2) External loads will have a beneficial effect on adaptation and fatigue, while adaptation and fatigue will fade over time, and the rate of decline is related to the individual's adaptation and fatigue level. To address the limitations of the traditional FFM, this study incorporates nonlinear dynamics to optimize the mathematical relationship between

adaptation and fatigue, differing from the first-order linear equations used in the original model [11,21]. This approach better reflects the real-world training response, accounting for the nonlinear nature of exercise training and the time-dependent fading of adaptation and fatigue.

(3) The change in performance level is reflected by the difference between adaptation and fatigue, a principle retained from the original FFM [11,20,19]. Additionally, this study introduces specificity coefficients to better capture individual differences in adaptation and fatigue dynamics, enabling the model to provide more personalized guidance based on individual characteristics.

Based on these modeling assumptions, the process of establishing an optimized mathematical model is as follows. After training, the regression coefficients of $A$ and $F$ are $\tau_a$ and $\tau_f$, respectively, namely:

$$\frac{dA}{dt} = -\tau_a A \tag{1}$$

$$\frac{dF}{dt} = -\tau_f F \tag{2}$$

$$(\tau_a > 0, \tau_f > 0)$$

The general solutions of equation (1) and (2) are the regression function of $A$ and $F$:

$$A = e^{-\tau_a t} + C_1 \tag{3}$$

$$F = e^{-\tau_f t} + C_2 \tag{4}$$

To further reflect individual differences, increasing the specificity coefficients $a$ and $f$:

$$A = a * e^{-\tau_a t} + C_1 \tag{5}$$

$$F = f * e^{-\tau_f t} + C_2 \tag{6}$$

The integration constants $C_1$ and $C_2$ were determined by estimating them alongside the model coefficients, based on the initial conditions (see Section 2.3.1 for estimation details). The new training load ($W_n$) will cause gains in A and F ($K_a$, $K_f$ is the gain coefficient), so on the nth day (n ≥ 1):

$$A_n = a * e^{-\tau_a t} + K_a W_n + C_1 \tag{7}$$

$$F_n = f * e^{-\tau_f t} + K_f W_n + C_2 \tag{8}$$

The performance level $P_n$ is:

$$P_n = A_n - F_n \tag{9}$$

$$P_n = \left(a * e^{-\tau_a t} + K_a W_n + C_1\right) - \left(f * e^{-\tau_f t} + K_f W_n + C_2\right) \tag{10}$$

$P_n$ represents the performance level on the nth day. $W_n$ represents the external load level on the nth day. $t$ represents time within the training period. $a$, $\tau_a$, $f$, and $\tau_f$ are parameters related to adaptation and fatigue regression. $K_a$ and $K_f$ are gain coefficients for adaptation and fatigue, respectively. $C_1$ and $C_2$ are constant parameters in the general solution of the function. The physiological interpretation of each parameter in Equations (10) is provided in S1 Table.

### 2.2 Data collection methods for model databases

**2.2.1 Subjects.** The subjects of this study are full-time university students in Beijing, China. The inclusion criteria are: ① Subjects have sufficient time; ② No other training arrangements during the experiment; ③ Volunteer to participate in the experiment and sign the informed consent form. Exclusion criteria are: ① Subjects have exercise contraindications; ② Have a history of physical or mental illness; ③ Simultaneously conducting high-dose training other than this experiment. Subjects' characteristics such as age, gender, BMI, and pre-experiment physical activity level (using the International Physical Activity Questionnaire long roll, a reliable and valid tool for physical activity assessment [38–40]) were collected before formal training.

The studies involving human participants were reviewed and approved by the Sport Science Experiment Ethics Committee of Beijing Sport University (No. 2022178H). The recruitment period is from September 1, 2022 to December 31, 2022. All methods were performed in accordance with relevant guidelines and regulations, and the research has obtained all subjects' written informed consent. This study did not include minors.

**2.2.2 Experimental design.** This study implemented 12 weeks of medium-intensity continuous cycling, with 3 sessions per week and 30 minutes per session. Prior to formal training, each participant underwent an incremental load cycling test to determine their individual $HR_{max}$. Based on this, the target heart rate range for training was set at 60–85% of each participant's $HR_{max}$ [41]. Before and after each training session, 5–10 minutes of warm-up or stretching activities were conducted, and each participant's training process was strictly monitored. Data were collected for each training session.

**2.2.3 Data collection methods.** In this study, the model input indicator is the external load collected during the each training session using an ergoselect100 power bicycle, a classic ergometer for exercise tests [42]. The model output indicator is the ratio of external load to internal load on the same day [43], which reflects the performance of the subject. To evaluate the internal load, HRV [35,44–48] and HRR related indicators [49–54] are selected for their ability to reflect autonomic nervous function, wearable compatibility, and potential for generalization in mass sports.

Data acquisition followed a specific timeline, as illustrated in Fig 1, divided into pre-training, training, and post-training phases. The data collection and indicators calculation methods are as follows.

(1) Pre-training

POLAR H10 heart rate band, a reliable and valid device for heart rate monitoring [55,56], was used to obtain the root mean square of the successive differences (RMSSD) of the difference between adjacent R-R intervals. The RMSSD was

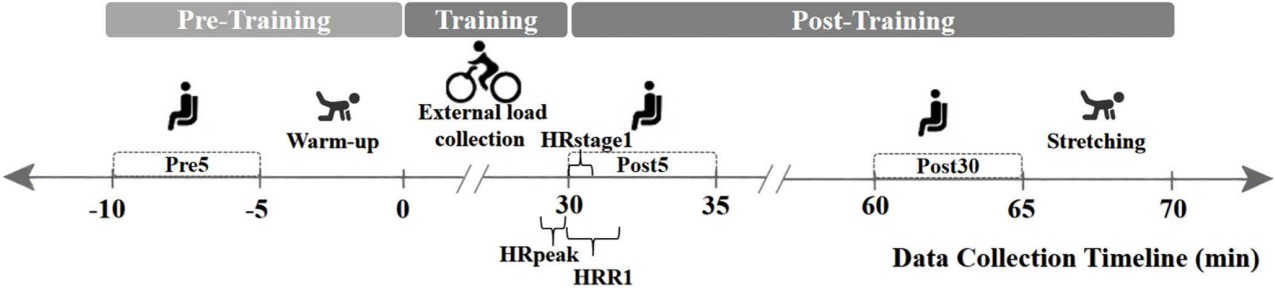

**Fig 1. Internal load indicator testing process.**

measured for 5 minutes while participants remained seated and inactive. This pre-training RMSSD data (Pre5) served as a baseline for HRV analysis. After the RMSSD test, participants were required to warm up before training.

(2) Training

During training sessions conducted on an ergoselect100 power bicycle [42], the bicycle speed and resistance level were continuously recorded (sampled at a frequency of 1 Hz). The external load was calculated by integrating the product of the speed and resistance level over time. Participants were required to reach the maximum value of the specified heart rate interval (85% $HR_{max}$) in the last minute of training to collect HRR indicators under submaximal intensity, denoted as $HR_{peak}$ in Fig 1.

(3) Post-training

Immediately after training, participants underwent a 5 minute recovery period while seated. Heart rate data was collected during this time to calculate post-training RMSSD (Post5) and HRR indicators.

The HRR indicators included the difference in HRR after 1 minute of training (ΔHRR1) and HRR percentage (HRr%), calculated using the formulas shown in equations (11) and (12):

$$\Delta HRR1 = HR_{peak} - HRR1 \tag{11}$$

$$HRr\% = \left( \frac{HR_{stage1} - HRR1}{HR_{stage1}} \right) \times 100\% \tag{12}$$

$HR_{peak}$: Peak heart rate in the last minute of training; HRR1: Average heart rate 1 minute after training; $HR_{stage1}$: Average heart rate 15 seconds after training.

Thirty minutes post-training, the RMSSD of participants was measured again while seated and resting for 5 minutes (Post30). This Post30 RMSSD data provided insights into longer-term autonomic nervous system recovery. Subsequently, the subjects were asked to perform muscle stretching.

The HRV index ($TL_{HRV}$) was calculated using Pre5, Post5, and Post30 RMSSD values for each training day to evaluate internal load, as shown in equation (13):

$$TL_{HRV} = \frac{Pre5 - Post5}{Post30 - Post5} \tag{13}$$

Pre5: 5 minutes RMSSD before formal training; Post5: 5 minutes RMSSD immediately after training; Post30: 30–35 minutes RMSSD after training.

## 2.3 Statistical analysis

Each indicator (external load and internal load) is normalized with an interval of 0.1–1 using MATLAB 2021a (MathWorks, USA). This specific normalization range prevents potential division by zero errors in the model output indicator calculation. The transformation maintained relative scale differences while ensuring mathematical validity across all observations.

### 2.3.1 Parameter estimation and fitting effect evaluation method.
Using the first 80% of training day data from each subject as the model learning database, model fitting and parameter estimation were performed using MATLAB 2021a (MathWorks, USA). Three internal load indicators (ΔHRR1, HRr%, $TL_{HRV}$) were used to calculate the output indicators of the model for fitting separately, as detailed in Section 2.2.3. These output indicators were then used in the cost function. The parameters were estimated by minimizing the sum of squared errors (SSE) between model-predicted values and observed values, using nonlinear least-squares minimization via the Trust-Region algorithm within the Curve

Fitting Toolbox. The optimization process utilized the toolbox's default configuration, including automatic initial parameter generation and standard convergence criteria. Initial conditions were handled by treating the integration constants $C_1$ and $C_2$ as adjustable parameters, which were estimated together with the other model coefficients ($a, \tau_a, K_a, f, \tau_f, K_f$). The fitted model surfaces were visualized as three-dimensional plots. The goodness of fit was evaluated using the coefficient of determination ($R^2$), SSE, and root mean square error (RMSE).

**2.3.2 Model prediction ability evaluation method.** Use the fitted parameters to predict the performance of each subject in the last 20% training day data (the model test database), compare it with the actual value, calculate the RMSE and mean absolute percentage error (MAPE) in the model test database, and evaluate the prediction ability of the model. Moreover, temporal dependency analysis using Spearman rank correlation analysis was used to investigate the relationship between prediction horizon (days since last training day in the model learning database) and absolute percentage error, assessing whether prediction errors systematically vary with temporal distance. Statistical significance was determined at $p < 0.05$.

To further evaluate the performance of the fitted model, the above indicators were compared with the same indicators' values obtained from the FFM original model before optimization [20,19]. The normality of differences was assessed using the Shapiro-Wilk test. Paired t-tests (for normally distributed differences) or Wilcoxon tests (for non-normally distributed differences) were then performed, with Bonferroni correction for multiple comparisons. All analyses were conducted using MATLAB 2021a (MathWorks, USA) at a significance level of $p < 0.05$. The original model is shown in equation (14):

$$P_n = A - F = K_a W_n - K_f W_n \tag{14}$$

$P_n$ represents the performance level on the nth day. $W_n$ represents the external load level on the nth day. $A$ and $F$ represents the adaptation and fatigue levels generated after training. $K_a$ and $K_f$ are gain coefficients for adaptation and fatigue, respectively.

## 3 Results

### 3.1 Subjects' characteristics and training completion status

Training data comprised paired external load and internal load measurements from each session, with 28–42 days of records per participant and a total of 420 paired data from 13 subjects. Due to the separate training of participants and the use of longitudinal data from each participant for personalized parameter estimation and model testing, the training completion status was different (Table 1). The Training Attendance Rate, defined as the ratio of actual training sessions attended to planned sessions and averaging 88.5±5.9% across all subjects, indicates excellent compliance among the participants. Most deviations from 100% attendance were attributed to menstrual cycles and the overlap of the final training week with the school winter break. Three overweight subjects (BMI ≥ 24 kg/m$^2$) and two subjects with low physical activity levels were given additional sports risk monitoring to prevent injury. None of the subjects withdrew from the study, further underscoring the good compliance throughout the research period. The detailed training data for each training session of all participants can be found in S1 File.

### 3.2 Fitting effect evaluation result

Using ΔHRR1, HRr%, and TL$_{HRV}$ to calculate the output indicators for model fitting and parameter estimation separately (the ratio of external load to internal load on the same day), each individual obtained their own three fitting models parameters using their longitudinal data (S2-S4 Tables).

When utilizing ΔHRR1 as a factor in calculating the output indicators of the model, the evaluation of the model fitting effect showed that all 13 subjects obtained small SSE and RMSE, with $R^2$ values ranging from 0.54 to 0.84 (54% of subjects exhibited $R^2$ values between 0.50 and 0.70, and 46% of subjects exhibited $R^2$ values between 0.70 and 0.90).

**Table 1. Subjects' characteristics and training situation.**

| Subjects number | Age | Gender | BMI (kg/m²) | Physical activity level | Total training period (days) | Actual training sessions (days) | Attendance Rate (%) |
|---|---|---|---|---|---|---|---|
| 1 | 20 | Female | 22.08 | Moderate | 72 | 31 | 86.1 |
| 2 | 23 | Female | 19.75 | High | 79 | 42 | 100.0 |
| 3 | 25 | Female | 23.80 | High | 73 | 32 | 88.9 |
| 4 | 25 | Female | 21.23 | Low | 71 | 31 | 86.1 |
| 5 | 25 | Male | 25.86 | Moderate | 79 | 34 | 94.4 |
| 6 | 18 | Female | 18.31 | High | 72 | 29 | 80.6 |
| 7 | 19 | Female | 21.01 | Moderate | 78 | 32 | 88.9 |
| 8 | 19 | Female | 22.87 | High | 76 | 33 | 91.7 |
| 9 | 19 | Female | 19.96 | Moderate | 75 | 28 | 77.8 |
| 10 | 19 | Male | 23.15 | High | 75 | 30 | 83.3 |
| 11 | 28 | Male | 31.43 | High | 72 | 32 | 88.8 |
| 12 | 19 | Male | 27.41 | High | 75 | 33 | 91.7 |
| 13 | 21 | Male | 22.99 | Low | 75 | 33 | 91.7 |
| **Summary Statistics** | **22±3** | **Male 38.5%, Female 61.5%** | **23.07±3.52** | **/** | **75±3** | **32±3** | **88.5±5.9** |

As for HRr%, the majority of subjects achieved good model fitting effect and low error, but there were also four subjects with $R^2$ values less than 0.50. When it comes to $TL_{HRV}$, most subjects achieved good fitting results, with $R^2$ values ranging from 0.40 to 0.95 and RMSE being low. However, three individuals had slightly high SSE. The fitting effect of the original model is significantly worse than that of the optimized fitting model in this study. Statistical analysis revealed significant increases in $R^2$ for ΔHRR1 (Bonferroni-corrected $p = 0.030$) and HRr% (Bonferroni-corrected $p = 0.021$). Moreover, most fitting indicators showed consistent positive trends in the optimized model, as presented in Fig 2. The SSE, RMSE, and $R^2$ values obtained through fitting with different output indicators are also shown in Fig 2. Specific values for each subject are provided in S5–S7 Tables.

### 3.3 Model prediction ability evaluation results

Although the prediction error of some subjects' fitting models is slightly higher than that of the original model, the overall predictive ability of the fitting model is still better. Temporal dependency analysis showed there were no significant relationships between prediction horizon and absolute percentage error for most subjects, indicating no population-wide consistency in temporal dependency of prediction accuracy. Significant but directionally inconsistent correlations were observed in a few cases, suggesting a highly individualized impact of temporal factors on prediction accuracy. The evaluation results of predictive ability are shown in Fig 3. Specific values for each subject, including those from the temporal dependency analysis, are provided in S8–S13 Tables.

Taking subject 4 as an example, the 3D images of the fitted model obtained using three output indicators are shown in Fig 4.

## 4 Discussion

The model optimized mathematical relationships in FFM, reflecting the time-varying characteristics of adaptation and fatigue. The fitting effect and predictive ability of the optimized fitting model are generally better than those of the original model, though individual variation exist. What's more, the use of longitudinal data also enables the model to have good fitting performance and predictive ability. HRV and HRR related indicators were used to verify the mathematical relationship in the model, and all indicators can bring good model fitting effects, indicating the generalizability of these indicators in mass sports. However, there are individual differences in the selection of the optimal output indicators of the model, and it



**Fig 2. R² (a), RMSE (b), and SSE (c) values obtained for each subject after fitting with different output indicators in the optimized and original model.** * indicates p < 0.05 (Bonferroni-corrected) where the optimized model significantly better. † indicates a non-significant trend favoring the optimized model (mean difference >0). †? indicates inconsistent direction.

is necessary to expand the collection of confounding factors, to explore and eliminate relevant factors that may introduce "noise" into the analysis.

The foundation for achieving good model performance is the selection of sensitive output indicators. Immediately after training, faster autonomic neuromodulation can lead to greater HRR in a shorter time, resulting in improved performance [49]. Because of that, the post-exercise HRR index which has previously served as a health monitoring indicator, has found widespread application in the training field [50–54] and is widely regarded as a robust indicator reflecting individual health and training status [49]. As for $TL_{HRV}$, HRV is sensitive to physiological and psychological stress, as well as steady-state disturbances, making it capable of immediately reflecting the training stimulus response after daily training, particularly moderate-intensity training [57]. Therefore, utilizing HRV as a guide for training is a reliable and promising method [35,44–48]. In summary, the three indicators selected in this study can effectively reflect the changes in response and performance after training, laying the foundation for the fitting of the model. What's more, these indicators can be conveniently collected through wearable devices, which are generalizable model output indicators that can be promoted in mass sports.

Moreover, the key to achieving good model performance in this study also lies in the application of longitudinal data to establish parameters for different individuals separately. To enhance the effectiveness of training, many scholars have

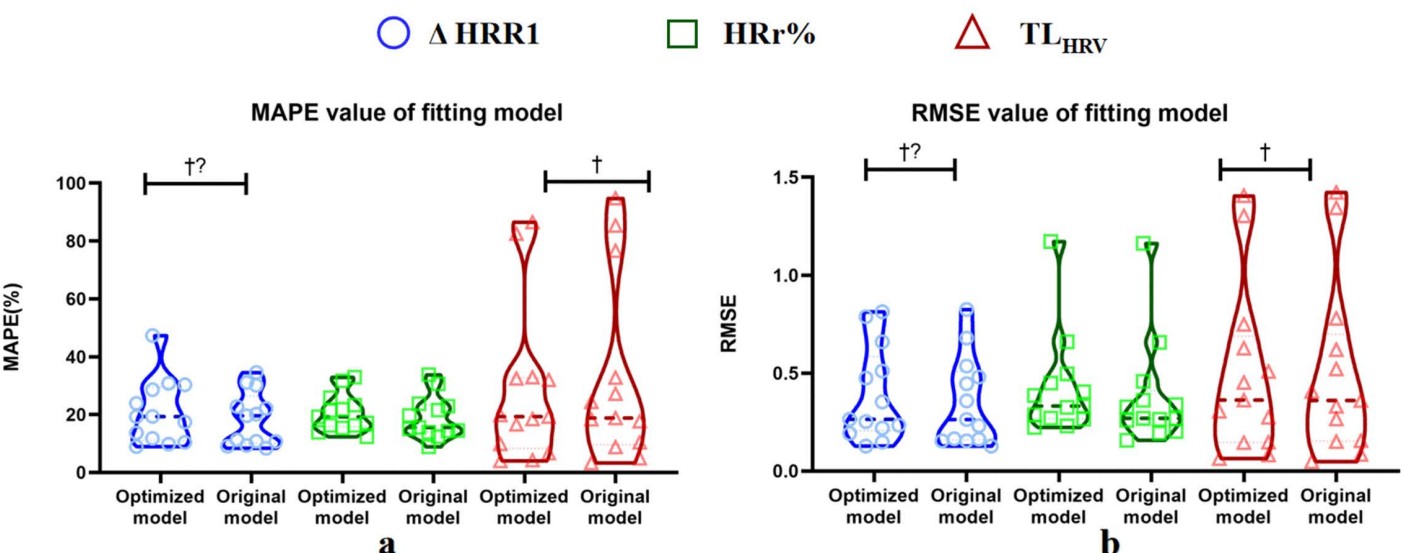

**Fig 3. MAPE (a) and RMSE (b) values obtained for each subject through model test database in the optimized and original model.** † indicates a non-significant trend favoring the optimized model (mean difference >0). †? indicates inconsistent direction.

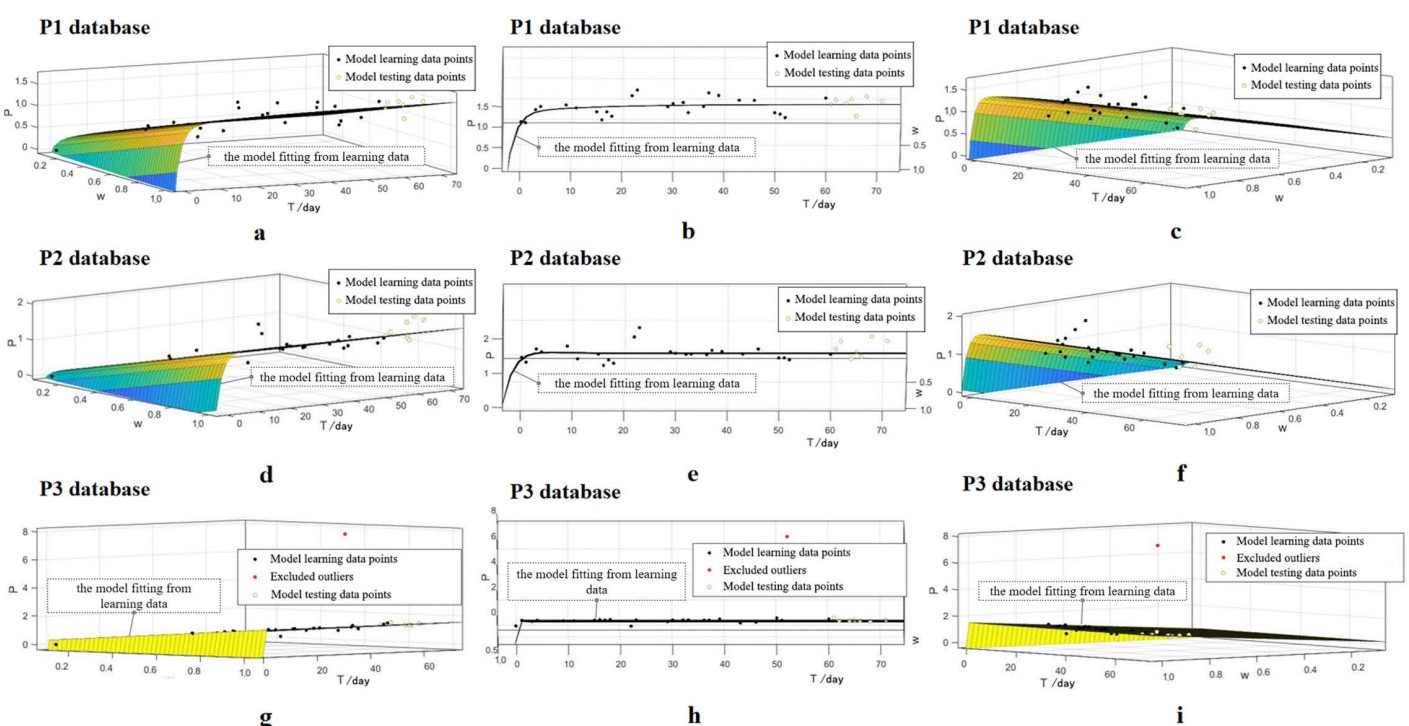

**Fig 4. Left side view(a, d, g), frontal view(b, e, h), and right side view(c, f, i) of the fitting models' 3D images of subject 4 using three databases separately.** The black dots represent the model learning data, the green dots represent the model testing data. P1, P2, and P3 represent model databases where the output indicators are calculated by HRR1, HRr%, and TL$_{HRV}$ respectively. W is the external load. P is the performance. T is the nth day within the training period.

pointed out the importance of using HRV and HRR indicators for longitudinal analysis. Some studies have indicated that when comparing different athletes horizontally, there is no consistent correlation between individuals with faster HRr% and better competition performance. However, in the longitudinal comparison of athletes themselves, changes in HRr% can better reflect post-exercise response and fatigue levels, thereby aiding in improved performance [58]. Similarly, HRV also has substantial individual differences, and many researchers recommend establishing longitudinal "HRV fingerprints" for individuals to monitor their HRV changes [58,59], which is consistent with this study. Marcelle investigated the role of HRV changes during the warm-up phase in training monitoring. They believe that HRV can effectively evaluate athletes' physiological status and has great potential in guiding training. However, individual differences exist, and further data is needed to validate the results of this monitoring approach in a larger and more homogeneous population [59]. Plews specifically emphasized the need to explore the HRV characteristics of athletes and implement longitudinal monitoring within each individual to track the unique "HRV fingerprints" of different athletes. Training should then be guided based on these fingerprints to enhance athletic performance. Multiple studies have also corroborated this perspective [34,45,58]. In collecting subjects' data, our study observed distinct personalized characteristics in the internal load of subjects, particularly in HRV indicators, where individuals exhibited significant variations in quiet HRV values. However, the HRV changes before and after training were remarkably similar, aligning with previous research findings [35,58]. Therefore, a significant breakthrough in training monitoring lies in identifying a method that can capture pre- and post-training indicator changes while also reflecting individual characteristics and identifying their unique "fingerprints" to enhance training guidance effectiveness. The model developed in this study successfully fulfills this requirement. By leveraging the characteristics of FFM, the model can mathematically depict the fundamental patterns before and after training. Additionally, the integration of machine learning empowers the model with the capability to identify an individual's "fingerprint" based on their training data, thereby facilitating personalized evaluation. Founded on real-time training data acquisition through wearable devices, the model incorporates machine learning to continuously learn and capture individual physiological characteristics. This approach establishes a foundation for developing adaptive training monitoring software [60]. Once integrated into smartphone applications, such software can provide users with an intuitive interface, offering personalized training adjustment recommendations, such as intensity modulation and rest advice. Moreover, temporal dependency analysis in this study further revealed that the temporal dependency of prediction errors lacks population-wide consistency, with only a few individuals showing significant but directionally inconsistent correlations. This suggests that the sensitivity of prediction accuracy to temporal distance is highly individualized. Future research could explore adding a "time-adaptive module" to exercise monitoring platforms. This module can determine whether an individual's model predictions exhibit temporal dependence based on their training data. For individuals with negative correlations, longer training cycles could be adopted, while those with positive correlations might benefit from high-frequency short cycles. This is crucial for enhancing the scientific nature of training in mass sports. Meanwhile, this application model also holds significant promise in both clinical and competitive sports fields. In clinical sports rehabilitation, customizing training loads according to individual recovery situations in a safer and more precise manner can maximize rehabilitation efficacy and minimize injury risks [61], which is consistent with the application outlook of optimizing the model in this study. As for competitive sports, given that the FFM originated in this domain, its combination with machine learning can further provide a basis for precise training.

Although three indicators have a good fitting effect, there is a "personalized" characteristic in the selection of the optimal output indicators of the model. When using HRr% to calculate the output indicators, subjects 2, 11, and 13 had slightly high fitting errors in their models, but they all achieved good fitting effects and predictive ability when using $\Delta HRR1$ and $TL_{HRV}$ as the output indicators. Similarly, the predictive ability test revealed that the majority of subjects exhibited good model fits with $TL_{HRV}$, while subjects 3 and 6 had slightly higher prediction errors. The above analysis suggests that there is a "personalized" characteristic in the selection of the optimal output indicators of the model. Individuals with varying levels of baseline training exhibit different degrees of changes in their autonomic nerve activities when subjected to the same intensity of training. Therefore, studying the patterns of HRR, HRV, and other indicators reflecting autonomic nerve

activities should be conducted on a population level. Several researchers have highlighted that individuals with different training levels display varying HRR responses [62], and factors such as training level and gender can influence the effectiveness of HRV-based endurance training guidance [45]. Consequently, the pre-training physical activity level and daily physical activity during training may lead to different autonomic nervous responses. Although this study required subjects to refrain from engaging in other training activities outside of the experiment, the pre-experiment physical activity levels were not screened, and daily physical activity was not restricted during the experiment. This could explain the slight variations in the optimal model output indicators observed among the participants in this study. Furthermore, even individuals with the same training level may exhibit subtle differences in their autonomic nervous responses, leading to specific changes in their HRR and HRV indicators. Some studies have highlighted that while both HRV and HRR can reflect changes in autonomic nerve activity, these two indicators have distinct physiological determinants. As a result, they may reflect different aspects of cardiac parasympathetic nervous system function [63]. Additionally, differences in the practical application of these two indicators have also been noted. Moreover, the specific factors that influence these indicators (such as training intensity, peripheral stimuli, genetic factors, etc.) are not yet fully understood, necessitating further exploration [35,45]. Therefore, the complexity of the physiological mechanisms underlying HRR and HRV may contribute to the varying optimal model output indicators observed among the participants in this study.

What's more, another possibility of the reasons that cause the "personalized" characteristic in the selection of the optimal output indicators is the "noise" in the data. Some experiments examining the autonomic nervous activity of subjects experiencing overtraining (particularly long-term overtraining) have reported different HRV responses among individuals, including increases, decreases, and no change [34]. Therefore, despite numerous practical studies demonstrating the effectiveness of HRV indicators in training monitoring [44,45,58,59], it is still crucial to exercise caution when interpreting their values [58]. Although the model developed in this study effectively captures changes in performance, it does not account for the occurrence of overtraining. Considering the complex physiological and biochemical reactions that can take place in the body after overtraining, the assumption of "no overtraining during the training process" was made when establishing the model in this study. Since this study solely utilizes collected data to test the model's fitting and predictive abilities, without employing the model for dynamic adjustment of the training plan, there may be deviations from this assumption during actual data collection, resulting in some "noise" that impacts the model's predictive performance. However, this is merely the author's speculation, and some researchers have also raised concerns regarding studies on overtraining individuals mentioned earlier. They argue that inconsistent results in such studies may be attributed to methodological differences, as the experimental design of these studies employs extreme training plans to induce an overtraining state, which does not align with real-world training scenarios [34]. Therefore, further investigation should be conducted to delve deeper into the changes and physiological mechanisms of output indicators under different conditions [35], to better understand the factors influencing model fitting and predictive effectiveness. Additionally, the differences in training period between participants may have influenced the model performance. Although the overall training attendance rate in our study was high and the missing training sessions did not show a regular impact on the model fitting effect, the variation in training period should not be overlooked. Longer training periods could provide more data points for parameter estimation, potentially enhancing model accuracy. However, due to individual differences in training responses, changes in training period do not directly correspond to proportional changes in model performance. Future studies could investigate this relationship further by controlling for training period or including it as a covariate in the analysis. To better capture the comprehensive impact of training period differences on model performance, it is also recommended that future research employ larger sample sizes and collect data over longer timeframes.

Overall, this study has several limitations that should be acknowledged. While the model demonstrated good fitting and prediction effects, it lacks an in-depth exploration of the personalized differences observed in various output indicators. Future research should focus on identifying optimal output indicators for individuals. This would provide a more robust foundation for selecting model indicators and achieving more stable fitting results. Additionally, the factors affecting model fitting

and prediction need to be studied further. Future studies should collect more confounding factors to explore and eliminate those that may introduce "noise" into the analysis. This will serve as a basis for further refining the model and enhancing its practical application. Moreover, the sample used in this study is homogeneous (average age 22±3 years) and only involves cycling as the training type. This limits the generalizability of the findings to other age groups and training types. Future research should validate the model across different sports, age groups, and genders to ensure its reliability and effectiveness in diverse populations. To address the risk of overfitting, future studies should also employ rigorous cross-validation techniques and consider incorporating additional variables that may influence model outcomes. Multicenter and randomized studies are also recommended to further establish the model's validity and applicability across different contexts. Finally, the current model does not account for overtraining. Future research could investigate extreme autonomic responses, such as paradoxical changes in HRV, to enhance the model's ability to detect and interpret specific training states.

## 5 Conclusion

The good performance of the model is attributed to the optimization of functional relationships and the application of individual longitudinal data. HRV and HRR related indicators are generalizable model output indicators that can be used to predict training effectiveness in mass sports, though individual differences in optimal indicators selection exist. This study provides a foundation for applying the model in wearable technology, offering a basis for developing adaptive training monitoring software. Such software, when integrated into smartphones, can provide users with personalized training recommendations, offering potential applications for general population, coaches, and rehabilitation physicians.

## Supporting information

**S1 Table. Physiological interpretation of optimized FFM parameters.**
(DOCX)

**S2 Table. Model parameter estimation results (using ΔHRR1 to calculate the output indicators).**
(DOCX)

**S3 Table. Model parameter estimation results (using HRr% to calculate the output indicators).**
(DOCX)

**S4 Table. Model parameter estimation results (using $TL_{HRV}$ to calculate the output indicators).**
(DOCX)

**S5 Table. Evaluation results of model fitting effect (using ΔHRR1 to calculate the output indicators).**
(DOCX)

**S6 Table Evaluation results of model fitting effect (using HRr% to calculate the output indicators).**
(DOCX)

**S7 Table. Evaluation results of model fitting effect (using $TL_{HRV}$ to calculate the output indicators).**
(DOCX)

**S8 Table. Evaluation results of model prediction ability and temporal dependency analysis (using ΔHRR1 to calculate the output indicators).**
(DOCX)

**S9 Table. The predicted values and actual values obtained from the model (using ΔHRR1 to calculate the output indicators and taking Subject 4 as an example).**
(DOCX)



**S10 Table. Evaluation results of model prediction ability and temporal dependency analysis (using HRr% to calculate the output indicators).**
(DOCX)

**S11 Table. The predicted values and actual values obtained from the model (using HRr% to calculate the output indicators and taking Subject 4 as an example).**
(DOCX)

**S12 Table. Evaluation results of model prediction ability and temporal dependency analysis (using TL$_{HRV}$ to calculate the output indicators).**
(DOCX)

**S13 Table. The predicted values and actual values obtained from the model (using TL$_{HRV}$ to calculate the output indicators and taking Subject 4 as an example).**
(DOCX)

**S1 File. Training database.**
(XLSX)

## Acknowledgments

We thank all the subjects who participated in this study. And we also acknowledge assistance from proof-readers and editors.

## Author contributions

**Conceptualization:** Wenxing Wang, Hong Ren.

**Data curation:** Wenxing Wang.

**Funding acquisition:** Wenxing Wang, Hong Ren.

**Investigation:** Wenxing Wang, Yuanhui Zhao.

**Methodology:** Wenxing Wang, Yuanhui Zhao, Xiao Hou, Wenlang Yu, Hong Ren.

**Supervision:** Hong Ren.

**Writing – original draft:** Wenxing Wang.

**Writing – review & editing:** Wenxing Wang, Yuanhui Zhao, Xiao Hou, Wenlang Yu, Hong Ren.

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
