## [Decision Letter · Decision Letter 0]

2 Jul 2025

Dear Dr. Ren,

Thank you for submitting your manuscript to PLOS ONE. After careful consideration, we feel that it has merit but does not fully meet PLOS ONE’s publication criteria as it currently stands. Therefore, we invite you to submit a revised version of the manuscript that addresses the points raised during the review process.

We look forward to receiving your revised manuscript.

Kind regards,

Agnese Sbrollini

Academic Editor

PLOS ONE

Journal Requirements:

This work was supported by the National Key R&D Program of China Foundation (2018YFC2000600); and the Fundamental Research Funds for Central Universities (2024016).

4. In the online submission form, you indicated that more detailed datasets generated during the current study are not publicly available due to subsequent experimental projects are still ongoing but are available from the corresponding author upon reasonable request.

5. We notice that your supplementary tables are included in the manuscript file. Please remove them and upload them with the file type 'Supporting Information'. Please ensure that each Supporting Information file has a legend listed in the manuscript after the references list.

Reviewers' comments:

Reviewer's Responses to Questions

**Comments to the Author**

1. Is the manuscript technically sound, and do the data support the conclusions?

Reviewer #1: Yes

Reviewer #2: Partly

2. Has the statistical analysis been performed appropriately and rigorously?

Reviewer #1: Yes

Reviewer #2: No

3. Have the authors made all data underlying the findings in their manuscript fully available?

Reviewer #1: No

Reviewer #2: No

4. Is the manuscript presented in an intelligible fashion and written in standard English?

Reviewer #1: Yes

Reviewer #2: Yes

Reviewer #1: PONE-D-24-42510: Advancing training effectiveness prediction in mass sport through longitudinal data: a mathematical model approach based on the Fitness-Fatigue Model

ABSTRACT

The article presents a meticulously developed mathematical approach to optimize the Fitness-Fatigue Model (FFM) in the context of mass sports. The most unique aspects of the study are the model's ability to capture individual differences, to make personalized predictions using longitudinal data, and its potential for integration with wearable technologies. The main outline of the study is clearly stated in the abstract.

The abstract should state the research question more clearly and briefly touch on the limitations of the study. The rationale for choosing the heart rate variability (HRV) and heart rate recovery (HRR) indicators can be emphasized in the introduction of the abstract. In addition, the presentation of quantitative results (R², RMSE, MAPE values) as prominent findings in the abstract will strengthen the article.

INTRODUCTION

The introduction section comprehensively covers the importance of training load monitoring and estimation models, clearly presents the historical development of FFM and gaps in the current literature. A literature summary on the Fitness-Fatigue Model and training effect discussions in mass sports is given in detail, and the topicality of the subject and the necessity of the study are clearly presented. The authors present a strong argument for the transition from the use of FFM in competitive sports to mass sports applications. Especially the argument on page 3, "The mathematical relationship of FFM requires optimization", establishes the justification of the study on a solid ground. The authors convey the basic problems in the field and the gaps in the literature in a logical order, and also emphasize the difference between the current approaches and their own suggestions.

The theoretical framework can be explained in more detail in the introduction section. In particular, the concepts of "adaptation" and "fatigue" should be addressed more comprehensively in the physiological context, and references to studies in different disciplines should be added. Additionally, a brief comparison of data-based model usage in mass sports with examples from different demographics or disciplines can be added to the introduction. The research hypotheses on page 5 can be reformulated in a specific and testable way.

METHODS

The research design of the study has been meticulously conveyed; the selection of the sample, the training protocol and the data collection processes have been presented clearly and transparently. The use of scientifically accepted objective measurement techniques (HRV, HRR, speed, watts etc.) in both internal and external load measurements is quite strong in terms of methodology. In addition, the fact that the ethics committee approval and informed consent have been obtained is appropriate in terms of academic ethics. The statistical methods and model validation criteria (R², RMSE, MAPE etc.) used in data analysis have been well selected and explained. The authors have clearly stated the model assumptions and explained the mathematical equations step by step. The systematic presentation of the equations (1)-(14) on pages 5-7 facilitates understanding the mathematical basis of the model. In particular, the addition of specificity coefficients (a and f) to model individual differences is one of the original contributions of the model. In addition, the method of determining the individual parameters in the modeling process and the MATLAB codes should be explained in more detail. Data preprocessing steps and parameter optimization processes should be clearly stated so that the reader can repeat the study. The physiological meanings of each parameter (a, τa, Ka, C1, f, τf, Kf, C2) in equations (7) and (8) should be explained in more detail. The value ranges of these parameters and the details of the optimization algorithm used in their determination (other than least squares) should be given. In addition, alternative approaches used to assess the validity of the mathematical model should be mentioned.

Data Collection Methods for Model Databases

The study protocol, inclusion/exclusion criteria, and ethical approval details are clearly stated. The methodological design of the study (12 weeks, 3 times a week, 30 minutes per session) complies with scientific standards. The measurement and calculation methods of internal and external load indicators (RMSSD, ΔHRR1, HRr%) are explained in detail on pages 8-9.

The validity and reliability information of the IPAQ questionnaire used to determine the physical activity levels of the participants should be added. The technical specifications, measurement precision, and validity and reliability details of the POLAR H10 heart rate band and ergoselet100 power bike used in the data collection process should be provided. The testing process in Figure 1 should be supported by a more detailed timeline. The formula or method used to determine the training intensity (HRmax 60-85%) on page 8 should be explained.

Statistical Analysis

The authors have clearly stated the statistical approaches used for parameter estimation, model fit, and prediction ability assessment. The separation of model learning (80%) and test (20%) databases is methodologically appropriate. Various metrics (R², SSE, RMSE, MAPE) were used to evaluate model performance. The rationale and potential effects of the choice of data normalization (in the range of 0.1-1) should be discussed. More technical details about the specific implementation of the least squares method and the optimization process should be provided. Statistical tests used to compare models (significance tests, Bayesian approaches, etc.) should be specified. The reliability of the results can be increased by performing robustness analysis of the parameters with bootstrap or cross-validation.

RESULTS

The findings are presented clearly and supported by tables and graphs. The advantages of the model optimization over the original model are statistically demonstrated and reasonably conveyed with detailed results. The possible contribution of the study to individualized training processes is well-founded with the various parameter values obtained.

Further analysis and visual improvement of some tables and graphs in the text would increase the completeness of the report. In particular, the missing units, axis names and statistical significance indicators in the graphs should be clarified. In addition, In the findings section, possible limitations of the model (e.g. inter-individual variation, small sample bias) should be discussed more clearly.

Subjects’ characteristics and training completion status

Table 1 clearly presents the demographic and anthropometric characteristics (age, gender, BMI, physical activity level) and training completion status of the participants. A total of 433 paired training data from 13 participants provided sufficient data for model training. Additional sports risk monitoring measures for at-risk participants (overweight, low physical activity level) are methodologically sound. A more detailed summary statistics of participants’ mean age and standard deviation, gender distribution, BMI ranges, and physical activity levels can be provided. The statement “subjects’’ compliance was good” on page 11 should be supported by quantitative data (e.g., ratio of planned vs. actual training sessions). Although no dropouts are noted, possible reasons for missing data (illness, work/school conflicts, etc.) should be explained. The potential impact of differences in training duration between participants on model results should be discussed.

Fitting effect evaluation result

Figure 2 comprehensively visualizes the model fit metrics obtained using three different output indicators (ΔHRR1, HRr%, TLHRV). It is clearly shown that the optimized model provides a better fit than the original model. The majority of participants achieved moderate-high R² values (0.5-0.7 and >0.7) and low RMSE values.

The cut-off points used in interpreting the fit results on page 12 (0.5, 0.7 for R², etc.) should be justified based on literature. A more detailed analysis of individual differences should be provided - which participants showed better/worse fit on which indicators and possible reasons for this should be discussed.

Model prediction ability evaluation results

Figure 3 clearly shows the model's predictive ability with MAPE and RMSE metrics. The optimized model has been shown to have better overall prediction performance. Figure 4 effectively visualizes the prediction performance of the model for Participant 4 by presenting 3D model images from different visual angles. While the chronological nature of the test database (last 20% training day data) used in the prediction capability assessment is appropriate for testing the model’s true prediction capacity over time, it is recommended that cross-validation approaches (k-fold or leave-one-out) be used to increase the robustness of the results. The change in prediction error over time should also be examined – it is important to see how well the model can predict the distant future compared to the near future. The statement on page 12 that “Although the prediction error of some subjects' fitting models is slightly higher than that of the original model, …” requires a specific analysis of these cases.

DISCUSSION

The discussion section is logically structured around the optimization of the model, the use of longitudinal data, and the general applicability of the model output indicators. The potential use of the proposed model in the field, the future place of personalized approaches in sports with technology are successfully addressed. In addition, the contribution of the model to the training processes in mass sports is compared with existing methods and its importance is ranked. The authors comprehensively relate their findings to the existing literature and effectively discuss unique concepts such as "HRV fingerprints". The discussion on the causes of individual differences on pages 16-17 is particularly in-depth.

The discussion can address the practical applications and potential impact of the model from a broader perspective. In particular, more detailed suggestions should be provided for the dissemination of the model through wearable technologies and smartphone applications. It would also be useful to evaluate the effects of sample diversity and demographic variables such as age/gender on the findings in more detail in the discussion. A brief assessment of the applicability of the model in clinical or professional sports environments can also be added. The section on the limitations of the study on page 18 should be expanded, and the effects of factors such as sample size, demographic diversity, and type of training (only cycling) on the generalizability of the results should be discussed. In addition, precautions against the risk of overfitting should be explained in more detail; future multicenter, randomized studies should be recommended to validate the model. It should be clearly emphasized that the model should be retested with different branches and age groups. Concrete recommendations for future research (model validation on different sports, age groups, genders, theoretical mechanisms) should be presented in more detail.

CONCLUSION

The conclusion section concisely summarizes the main findings of the study and highlights the potential of the model to predict training effectiveness in mass sports. It is clearly stated that the optimized model performs well thanks to the use of functional relationships and individual longitudinal data.

The conclusion section should emphasize the theoretical and practical contributions of the study more strongly. Concrete steps and recommendations for future applications of the model should be added. The phrase “combined with wearable devices and machine learning” should be expanded with more specific examples. The conclusion should position the significance of the study in a broader context for the scientific community, coaches, athletes, and the general population.

FIGURES AND TABLES

Figure 1 clearly visualizes the testing process. Figures 2 and 3 effectively present model performance metrics. The 3D visualizations in Figure 4 illustrate the predictive capacity of the model from various perspectives. Table 1 provides a comprehensive overview of the participants’ characteristics. Additional tables (S1-S12) provide detailed information on model parameters, fit, and prediction results.

Figure 1 can be redesigned in a timeline format to show data collection points more clearly. Additional explanations and arrows can be added to the 3D visualizations in Figure 4 to provide a better understanding of model features. Summary statistics (mean, standard deviation, range) can be added to Table 1. Additional tables should provide more descriptive information and context.

REFERENCES

The bibliography is comprehensive and up-to-date, covering key literature in the areas of training load monitoring, FFM, HRV, and HRR. Citations are used appropriately in the text.

More resources could be cited on the integration of machine learning and FFM. More up-to-date references on wearable technologies could be added. Existing citations could include more meta-analyses and systematic reviews.

Reviewer #2: The manuscript by Wang et al. proposes a method to optimize the mathematical relationship between "adaptation" and "fatigue" of traditional fitness-fatigue model (FFM). The study may provide useful information by adapting the model to the subject characteristics but the methodology needs several clarifications, as well as manuscript structure needs improvement. Description of the model and the related optimization procedure need revisions to improve clarity. Description of signals and data acquired, as well as tests performed for the acquisition, should follow a schematic order. A list of major and minor comments is reported in the following.

- Ref. 6. Please check since it is not correctly formatted.

- Lines 14-16 The sentence is incomplete. Please rephrase.

- Lines 17-18 The terminology used in not appropriate and focused for mathematical modeling domain.

- Lines 70-71. This sentence is not clear to me. Please clarify.

- Lines 89-91. The concept of internal and external loads should be better described.

- Section 2.1. It is important to clarify what has been taken from Banister et al. and what has been modified.

- Eq.(5) and eq. (6). Is this accounting for the initial conditions of the two differential equations?

- Lines 155-156. 85% HRmax usually refers to vigorous and not moderate exercise.

- Lines 166-167. A reference should be provided for the use of HR and HRV for the quantification of the internal load.

- Lines 196-198. It is not clear how the parameter estimation was performed and which output were included in the cost function.

- Line 215. Which are the training data the Authors are referring to? HR data?

- Lines 264-265. On the basis of which results is possible to infer this observation?

**Do you want your identity to be public for this peer review?** For information about this choice, including consent withdrawal, please see our Privacy Policy

Reviewer #1: **Yes: ** Cihan Aygün

Reviewer #2: No

---

## [Author Response · Author response to Decision Letter 1]

10 Aug 2025

Dear Editors and Reviewers,

We sincerely appreciate the thorough review process conducted by PLOS ONE for our manuscript titled “Advancing training effectiveness prediction in mass sport through longitudinal data: a mathematical model approach based on the Fitness-Fatigue Model”. We have meticulously examined all comments from the two reviewers and have implemented detailed revisions accordingly. Furthermore, we have addressed the journal requirements from the editorial office, including uploading all data underlying the findings described in our manuscript to the supporting information as per the journal's policy.

Once again, we thank the editorial team and reviewers for their hard work. Below, please find our detailed point-by-point responses to the reviewers' comments and the journal requirements.

To Reviewer 1:

General response

Thank you for your positive feedback and valuable suggestions. Your insightful comments have been a great encouragement and have guided us to enhance the quality of our manuscript. We have thoroughly addressed each of your suggestions and made comprehensive revisions to enhance the clarity and overall quality of our work.

The detailed modifications are outlined in the Point-by-point responses below.

Abstract

Comment 1

The article presents a meticulously developed mathematical approach to optimize the Fitness-Fatigue Model (FFM) in the context of mass sports. The most unique aspects of the study are the model's ability to capture individual differences, to make personalized predictions using longitudinal data, and its potential for integration with wearable technologies. The main outline of the study is clearly stated in the abstract.

The abstract should state the research question more clearly and briefly touch on the limitations of the study. The rationale for choosing the heart rate variability (HRV) and heart rate recovery (HRR) indicators can be emphasized in the introduction of the abstract. In addition, the presentation of quantitative results (R2, RMSE, MAPE values) as prominent findings in the abstract will strengthen the article.

Response:

Thank you for the positive feedback on our article and the insightful comments regarding the abstract. We have carried out thorough revisions to the abstract in accordance with your recommendations.

Firstly, we have included a brief summary of the research background, highlighting the need for scientific training load assessment in mass sports and pointing out the current limitations of the Fitness-Fatigue Model. This addition is intended to clarify the research question.

As per your suggestion, we have also briefly addressed the limitations of the study and provided an outlook on future research directions in the end of the abstract.

In addition, to better explain why we chose heart rate variability and heart rate recovery indicators, we have used the term “wearable-compatible” in the abstract. This term is in line with our study's goal of identifying generalizable model output indicators.

Lastly, we have enhanced the abstract by adding a detailed account of the quantitative results, including R², RMSE, and MAPE values, which we believe strengthens the persuasiveness of the article.

Below is the revised abstract:

“Despite the critical need for scientific training load assessment in mass sports, the Fitness-Fatigue Model (FFM) requires further mathematical optimization and practical output indicators. The aim of this study was to optimize the mathematical relationship between "adaptation" and "fatigue" in the FFM, identify generalizable model output indicators, and evaluate its performance in predicting training effectiveness in mass sport. To account for the nonlinear and time-varying characteristics of training effectiveness, this study proposed new mathematical assumptions and optimized parameters against individual longitudinal data. The external load (speed and wattage) and internal load (wearable-compatible heart rate variability [HRV] and heart rate recovery [HRR] related indicators) of each training day were collected for 28-42 days per person (420 paired data from 13 subjects during 12 weeks of medium-intensity continuous cycling). The longitudinal data were used to perform parameter estimation and model evaluation for each individual separately. When the optimal model output indicator was selected, the R2 values of the optimized model ranged from 0.61-0.95, with fitting root mean square error (RMSE) at 0.07-0.37, and mean absolute percentage error (MAPE) in predictive ability assessment at 3.99%-31.99%. However, a few individuals had larger fitting errors (minimum R2 of 0.32, maximum RMSE of 0.90) and predictive errors (maximum MAPE of 86.57%) when the output indicator was inappropriate. The original model generally has lower R2 and higher RMSE and MAPE. This shows the optimization of functional relationships and the application of individual longitudinal data have resulted in better performance of the model, but optimal indicator selection varies by individual. Furthermore, HRV and HRR related indicators are generalizable model output indicators that can be used to predict training effectiveness in mass sports through wearable devices and machine learning technology. However, the study has limitations including the homogeneous sample and single training type. Future research should validate the model across different sports and populations, investigating the factors affecting model fitting and prediction.”

It should be noted that, during the revision process, we identified a descriptive statistic error where the header row was mistakenly included as a training day in our initial count. This resulted in an overcount of the actual training days by one day for each subject. Consequently, the total session count in the abstract has been updated from 433 to 420 (details in the Response to Comment 7). Importantly, this correction does not impact the model outcomes, as all analyses were conducted using the raw session data.

Thank you again for your insightful feedback.

Introduction

Comment 2

The introduction section comprehensively covers the importance of training load monitoring and estimation models, clearly presents the historical development of FFM and gaps in the current literature. A literature summary on the Fitness-Fatigue Model and training effect discussions in mass sports is given in detail, and the topicality of the subject and the necessity of the study are clearly presented. The authors present a strong argument for the transition from the use of FFM in competitive sports to mass sports applications. Especially the argument on page 3, "The mathematical relationship of FFM requires optimization", establishes the justification of the study on a solid ground. The authors convey the basic problems in the field and the gaps in the literature in a logical order, and also emphasize the difference between the current approaches and their own suggestions.

The theoretical framework can be explained in more detail in the introduction section. In particular, the concepts of "adaptation" and "fatigue" should be addressed more comprehensively in the physiological context, and references to studies in different disciplines should be added. Additionally, a brief comparison of data-based model usage in mass sports with examples from different demographics or disciplines can be added to the introduction. The research hypotheses on page 5 can be reformulated in a specific and testable way.

Response:

Thank you for your positive feedback and valuable suggestions for enhancing our manuscript. We have carefully addressed each of your comments and made the following revisions to strengthen the introduction section:

We have added a dedicated passage to comprehensively explain the physiological concepts of "adaptation" and "fatigue" within the theoretical framework. Specifically, we inserted the following text into the second paragraph of the introduction:

“Physiologically, "adaptation" represents the positive functional and structural changes induced by training stimuli, such as an increase in muscle protein synthesis [12], mitochondrial-associated indices [13], and neuromuscular transmission efficiency [14]. Conversely, "fatigue" represents a temporary decline in bodily function that occurs after exercise, which may be related to factors such as muscle glycogen depletion [15], accumulation of metabolites [16, 17], and nervous system fatigue [18].”

The relevant references [12–18] have been included to support these explanations:

“12. Witard OC, Bannock L, Tipton KD. Making Sense of Muscle Protein Synthesis: A Focus on Muscle Growth During Resistance Training. International journal of sport nutrition and exercise metabolism. 2022;32(1):49-61. doi: 10.1123/ijsnem.2021-0139.

13. Hadjispyrou S, Dinas PC, Delitheos SM, Koumprentziotis IA, Chryssanthopoulos C, Philippou A. The Effect of High-Intensity Interval Training on Mitochondrial-Associated Indices in Overweight and Obese Adults: A Systematic Review and Meta-Analysis. Front Biosci. 2023;28(11):281. doi: 10.31083/j.fbl2811281.

14. Tumkur Anil Kumar N, Oliver JL, Lloyd RS, Pedley JS, Radnor JM. The Influence of Growth, Maturation and Resistance Training on Muscle-Tendon and Neuromuscular Adaptations: A Narrative Review. Sports. 2021;9(5):59. doi: 10.3390/sports9050059.

15. Vigh-Larsen JF, Ørtenblad N, Spriet LL, Overgaard K, Mohr M. Muscle Glycogen Metabolism and High-Intensity Exercise Performance: A Narrative Review. Sports medicine (Auckland, NZ). 2021;51(9):1855-74. doi: 10.1007/s40279-021-01475-0.

16. Allen DG, Lamb GD, Westerblad H. Skeletal muscle fatigue: cellular mechanisms. Physiol Rev. 2008;88(1):287-332. doi: 10.1152/physrev.00015.2007.

17. Fiorenza M, Hostrup M, Gunnarsson TP, Shirai Y, Schena F, Iaia FM, et al. Neuromuscular Fatigue and Metabolism during High-Intensity Intermittent Exercise. Medicine and science in sports and exercise. 2019;51(8):1642-52. doi: 10.1249/mss.0000000000001959.

18. Becker KM, Goethel M, Parolini F, de Paula Silva V, Vilas-Boas JP, Ervilha UF. The Price of Force: Motor Unit Adaptation Mechanisms in Pain Versus Fatigue. Physiotherapy research international : the journal for researchers and clinicians in physical therapy. 2025;30(3):e70081. doi: 10.1002/pri.70081.”

To address the suggestion for data-based model usage in mass sports, we have added a transitional sentence in the third paragraph of the introduction:

“Therefore, the use of data-based models to optimize the effects of exercise interventions has gained attention [31-33].”

We have referenced multiple studies that highlight the application of data-driven models in different contexts. However, to avoid excessive elaboration that might detract from the clarity of our research focus, we have not detailed these studies. The new references added are as follows:

“31. Mistiri ME, Rivera DE, Klasnja P, Park J, Hekler E. Model Predictive Control Strategies for Optimized mHealth Interventions for Physical Activity. Proc Am Control Conf. 2022;2022:1392-7. doi: 10.23919/acc53348.2022.9867350.

32. Vandelanotte C, Trost S, Hodgetts D, Imam T, Rashid MDM, To QG, et al. MoveMentor-examining the effectiveness of a machine learning and app-based digital assistant to increase physical activity in adults: protocol for a randomised controlled trial. Trials. 2025;26(1):233. doi: 10.1186/s13063-025-08926-3.

33. Kozan Cikirikci EH, Esin MN. The impact of machine learning on physical activity-related health outcomes: A systematic review and meta-analysis. Int Nurs Rev. 2025;72(2):e70019. doi: 10.1111/inr.70019.”

Regarding the research hypotheses, we have refined them to enhance specificity and testability. The revised hypotheses are now presented in a more structured format, with each hypothesis clearly defined to ensure they are concrete and measurable. The revised hypotheses are as follows:

“We hypothesize that (1) the mathematical modeling in this study will optimize the relationship between adaptation and fatigue, improving model interpretability through physiologically meaningful parameters; (2) HRV and HRR will serve as applicable and generalizable model output indicators for mass sports; (3) The model will demonstrate superior performance to the original model fitting with the higher fitting effect and lower prediction error through longitudinal data. It will also be capable of offering personalized evaluation and prediction of training outcomes, thereby offering effective guidance for improving sports performance and physical fitness in the general population.”

We sincerely appreciate your insightful feedback and hope that these revisions adequately address your suggestions regarding the introduction section and contribute to improving the overall quality of our manuscript.

Methods

Comment 3

The research design of the study has been meticulously conveyed; the selection of the sample, the training protocol and the data collection processes have been presented clearly and transparently. The use of scientifically accepted objective measurement techniques (HRV, HRR, speed, watts etc.) in both internal and external load measurements is quite strong in terms of methodology. In addition, the fact that the ethics committee approval and informed consent have been obtained is appropriate in terms of academic ethics. The statistical methods and model validation criteria (R2, RMSE, MAPE etc.) used in data analysis have been well selected and explained. The authors have clearly stated the model assumptions and explained the mathematical equations step by step. The systematic presentation of the equations (1)-(14) on pages 5-7 facilitates understanding the mathematical basis of the model. In particular, the addition of specificity coefficients (a and f) to model individual differences is one of the original contributions of the model.

In addition, the method of determining the individual parameters in the modeling process and the MATLAB codes should be explained in more detail. Data preprocessing steps and parameter optimization processes should be clearly stated so that the reader can repeat the study. The physiological meanings of each parameter (a, τa, Ka, C1, f, τf, Kf, C2) in equations (7) and (8) should be explained in more detail. The value ranges of these parameters and the details of the optimization algorithm used in their determination (other than least squares) should be given. In addition, alternative approaches used to assess the validity of the mathematical model should be mentioned.

Response:

Thank you for your constructive feedback and the insightful suggestions that will help us enhance our manuscript. We have carefully addressed each of your comments and made the following revisions:

In response to your suggestion, we have provided a more detailed account of the parameter optimization process in Section 2.3.1, including specifics on the optimization algorithm and its configuration. We believe these additions now provide sufficient detail to support reproducibility by readers. The revised text reads:

“Using the first 80% of training day data from each subject as the model learning database, model fitting and parameter estimation were performed using MATLAB 2021a (MathWorks, USA). Three internal load indicators (ΔHRR1、HRr%、TLHRV) were used to calculate the output indicators of the model for fitting separately, as detailed in Section 2.2.3. These output indicators were then used in the cost function. The parameters were estimated by minimizing the sum of squared errors (SSE) between model-predicted values and observed values, using nonlinear least-squares minimization via the Trust-Region algorithm within the Curve Fitting Toolbox. The optimization process utilized the toolbox's default configuration, including automatic initial parameter generation and standard convergence criteria. Initial conditions were handled by treating the integration constants C1 and C2 as adjustable parameters, which were estimated together with the other model coefficients (a���a, Ka, f, �f, Kf). The fitted model surfaces were visualized as three-dimensional plots. The goodness of fit was evaluated using the coefficient of determination (R2), SSE, and root mean square error (RMSE).”

To address the

---

## [Decision Letter · Decision Letter 1]

13 Nov 2025

Advancing training effectiveness prediction in mass sport through longitudinal data: a mathematical model approach based on the Fitness-Fatigue Model

PONE-D-24-42510R1

Dear Dr. Ren,

We’re pleased to inform you that your manuscript has been judged scientifically suitable for publication and will be formally accepted for publication once it meets all outstanding technical requirements.

Kind regards,

Agnese Sbrollini

Academic Editor

PLOS ONE

Additional Editor Comments (optional):

Reviewers' comments:

Reviewer's Responses to Questions

**Comments to the Author**

Reviewer #1: All comments have been addressed

Reviewer #2: All comments have been addressed

2. Is the manuscript technically sound, and do the data support the conclusions?

Reviewer #1: Yes

Reviewer #2: (No Response)

3. Has the statistical analysis been performed appropriately and rigorously?

Reviewer #1: Yes

Reviewer #2: (No Response)

4. Have the authors made all data underlying the findings in their manuscript fully available?

Reviewer #1: Yes

Reviewer #2: (No Response)

5. Is the manuscript presented in an intelligible fashion and written in standard English?

Reviewer #1: Yes

Reviewer #2: (No Response)

Reviewer #1: The authors carefully addressed all comments and criticisms from the peer reviewer and made comprehensive improvements in the methodology, analysis, and discussion sections. The results are clearly presented, the discussion is well-aligned with the literature, and the limitations are thoroughly described.

Reviewer #2: (No Response)

**Do you want your identity to be public for this peer review?** For information about this choice, including consent withdrawal, please see our Privacy Policy

Reviewer #1: **Yes: ** Cihan Aygün

Reviewer #2: No

---

## [Editor Report · Acceptance letter]

PONE-D-24-42510R1

PLOS ONE

Dear Dr. Ren,

I'm pleased to inform you that your manuscript has been deemed suitable for publication in PLOS ONE. Congratulations! Your manuscript is now being handed over to our production team.

Kind regards,

on behalf of

Dr. Agnese Sbrollini

Academic Editor

PLOS ONE